# RFFNet: Towards Robust and Flexible Fusion for Low-Light Image Denoising

## ABSTRACT

Low-light environments will introduce high-intensity noise into images. Containing fine details with reduced noise, near-infrared/flash images can serve as guidance to facilitate noise removal. However, existing fusion-based methods fail to effectively suppress artifacts caused by inconsistency between guidance/noisy image pairs and do not fully excavate the useful information contained in guidance images. In this paper, we propose a robust and flexible fusion network (RFFNet) for low-light image denoising. Specifically, we present a multi-scale inconsistency calibration module to address inconsistency before fusion by first mapping the guidance features to multi-scale spaces and calibrating them with the aid of pre-denoising features in a coarse-to-fine manner. Furthermore, we develop a dual-domain adaptive fusion module to adaptively extract useful high-/low-frequency signals from the guidance features and then highlight the informative frequencies. Extensive experimental results demonstrate that our method achieves state-of-the-art performance on NIR-guided RGB image denoising and flash-guided no-flash image denoising.

## CCS CONCEPTS

• **Computing methodologies** → **Computer vision**; **Image manipulation**.

## KEYWORDS

Deep learning, Image fusion, Image denoising

## 1 INTRODUCTION

Images captured in low-light environments typically contain distracting noise. Low-light image denoising aims to eliminate undesired noise, playing essential roles in all-day surveillance, self-driving techniques, and photography. Image denoisers taking a single image as input often results in insufficient denoising or over-smoothing when dealing with these heavily degraded noisy images.

Guided image restoration [11, 18, 22, 32, 34, 36] provides a new solution for image denoising. With low noise, clear structure, and rich details, near-infrared (NIR) images and flash images can serve as guidance for low-light image denoising. Nonetheless, there exists inconsistency between target and guidance image pairs, such as the presence of highlights on objects and hard shadows behind objects under NIR or flash illumination. As illustrated in the bottom part of

*MM '24, 28 October - 1 November, 2024, Melbourne, Australia*

© 2024 Copyright held by the owner/author(s). Publication rights licensed to ACM.
ACM ISBN 978-1-4503-XXXX-X/18/06
https://doi.org/XXXXXXX.XXXXXXX

Fig. 1 (b), the shadows present in the NIR image do not exist in the RGB image. As a result, directly fusing cross-domain information tends to introduce artifacts, which makes low-light guided denoising a complex problem, as it requires not only removing strong noise and restoring details but also suppressing artifacts.

Recently, several works have achieved impressive performance in NIR-guided [21, 35] and flash-guided [11, 31, 39] image denoising based on elaborately designed fusion methods. Nevertheless, they still struggle to eliminate the inconsistency issues. For example, CUNet [11] learns the common and unique features between the noisy image and the guidance image but fails under strong noise, leading to insufficient denoising and the production of artifacts, as illustrated in Fig. 1 (d). SANet [35] guides denoising by aggregating structural signals from the guidance image, but introduces inconsistent structures, resulting in artifacts in the output (see Fig. 1 (e)). Moreover, kernel prediction-based methods [31, 39] are highly susceptible to noise, failing to effectively suppress the artifacts in the presence of strong noise.

Additionally, existing methods do not fully excavate the useful information contained in the guidance images in terms of frequency. For instance, DVN [21] leverages an additional branch to incorporate only the structure of guidance images, disregarding the potential beneficial information in low-frequency, such as smooth transition. SANet aggregates structure without considering the roles of individual frequency components, causing low-frequency colors to be equally integrated into the structure map, resulting in unexpected color deviations. We posit that high-frequency information can guide the restoration of texture details, while low-frequency signals are useful for noise removal by prompting spatial smoothness in flat areas.

To alleviate the above-mentioned issues, this paper proposes a Multi-Scale Inconsistency Calibration Module (MICM) to handle the inconsistency between the input image pairs. Specifically, we downsample the features from the input image pair into different scales to decouple noise and structure, and then modulate the cross-spectral features with the aid of pre-denoised features from the noisy input using spatial attention. As a result, the inconsistency can be handled in a coarse-to-fine manner. Furthermore, we develop a Dual-Domain Adaptive Fusion Module (DAFM) to flexibly inject the modulated guidance features into the main path of the network by considering different roles of low-/high-frequency signals. More concretely, we employ multi-kernel depthwise convolutions and average pooling techniques to extract high-frequency and low-frequency components of cross-modal images. Then, squeeze-and-excitation techniques are utilized to extract relevant local contexts from the high-frequencies of the guidance image, while useful global contexts in low-frequencies of the guidance image are captured through channel-wise cross-attention.

As illustrated in Fig. 1, our RFFNet strikes a better balance in noise removal, details recovery, and artifact suppression based on

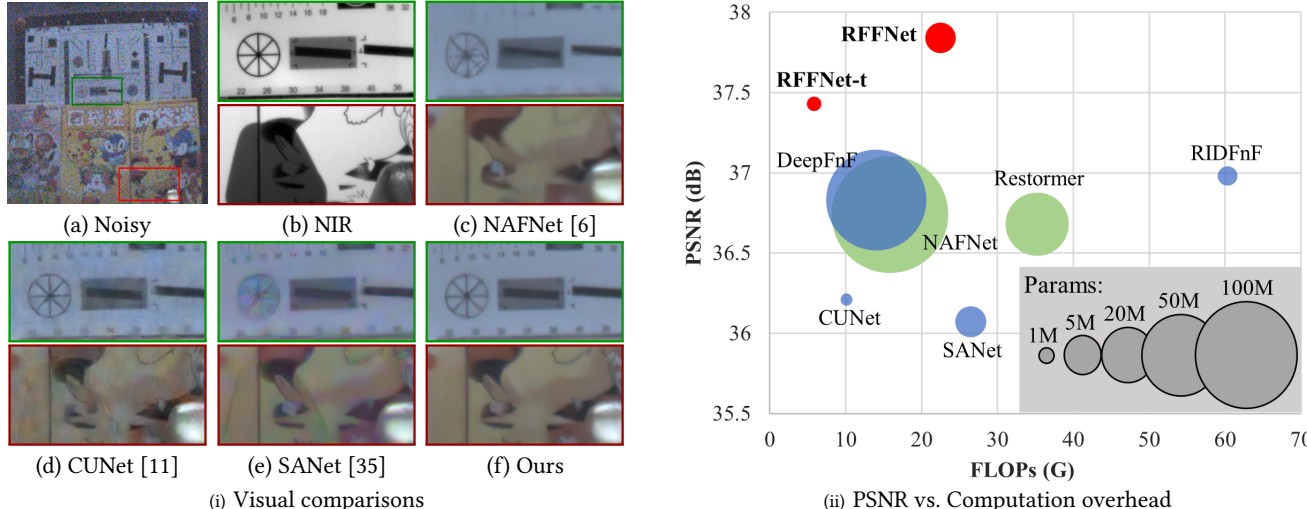

**Figure 1: Superiority of our method. (a) Visual comparisons on a challenging noisy RGB-NIR image pair. Our method produces a better denoising result with clear details and fewer artifacts. (b) PSNR vs. Computation overhead on FAID [39]. Our method achieves the superior performance while maintaining efficiency.**

robust and flexible fusion, while maintaining efficiency. Overall, the main contributions of this study are as follows:

- We propose a Multi-Scale Inconsistency Calibration Module that decouples noise and structure by transforming cross-modal features into multi-scale scales, pursuing a coarse-to-fine manner to address the local inconsistency between noisy and guidance features.
- We develop a Dual-Domain Adaptive Fusion Module, which selectively extracts the high-frequency and low-frequency components from the guidance features and restores the salient structure and flat areas differentially.
- We propose a general guided image denoising formulation in a calibration-before-fusion manner by integrating MICM and DAFM into a two-stage backbone. The proposed algorithm achieves state-of-the-art performance in both NIR/flash-guided image denoising in low-light environments.

## 2 RELATED WORKS

### 2.1 Single-Image Denoising

Traditional single image denoising algorithms mainly include filtering [3, 10], sparse-coding [14, 41] and low-rank factorization [13, 17], which often rely on handcrafted filters or prior knowledge and lack flexibility in handling complex noise distributions.

In recent years, learning-based methods have outperformed traditional methods. For example, DnCNN [45] uses batch normalization and residual learning for denoising. RIDNet [2] introduces feature attention in the residual structure to remove real noise. SADNet [4] employs deformable convolutions for spatially adaptive denoising. NBNet [7] removes noise by learning a set of reconstruction bases in the feature space. MPRNet [44] builds a multi-stage network for progressive restoration. NAFNet [6] designs a novel attention block to build a hierarchical network. Transformer-based denoising networks, such as Uformer [38] and Restormer [43], are developed

to better use local contexts and long-range dependencies. However, single-image denoising approaches tend to over-smooth texture details, especially at high noise levels.

### 2.2 Guided Image Restoration

Several early works on guided image restoration [15, 25, 33] utilize flash images to reduce noise and blurring artifacts in non-flash images. Afterwards, He et al. [18] propose guided filters to handle various low-level tasks, including flash-guided no-flash image denoising. Yan et al. [42] generate a scale map to capture smooth transitions and usable edges in guidance images.

Recently, deep learning methods have been applied to guided image restoration. DJFR [28] constructs a general deep fusion network for multiple image processing tasks, including image denoising. SVLRM [32] proposes a spatially variant linear representation model with learnable coefficients. Subsequently, CUNet [11] uses sparse encoding to separate the common and unique information of cross-modal images, which can perform both guided restoration and guided fusion. DKN [22] learns explicitly sparse and spatially variant kernels to guide filtering. DeepFnF [39] and RIDFnF [31] predict the kernel to combine the pixel colors of flash/no-flash images. DVN [21] addresses the issue of structure inconsistency in NIR images by using deep inconsistency prior. SANet [35] aggregates structure from the guidance image to estimate a clean structure map for guided denoising. However, when dealing with dense noise in low-light environments, most existing guided denoising methods fail to strike a balance between detail preservation, noise removal, and artifact suppression.

### 2.3 Frequency-Relevant Image Restoration

Frequency analyses are widely adopted in image restoration [8]. Traditional algorithms for denoising [10, 20] or deblurring [26, 27]

design adaptive filters in the frequency domain. Many learning-based image restoration methods are approached from a frequency perspective [8]. MWCNN [29] replaces the conventional pooling operation with the wavelet transform for a better tradeoff between receptive field sizes and efficiency. In [16], stochastic frequency masking is proposed to address overfitting issues in blind super-resolution and blind denoising. SDWNet [47] designs a wavelet transformation module to recover clear high-frequency texture details. FADN [40] decouples input images into multiple components using discrete cosine transform for individual processing, enabling differential super-resolution. SFNet [9] proposes a decoupling and modulation module that can adaptively separate frequency information and selectively aggregate them. This paper emphasizes the diverse roles of frequencies in guidance images and strives to adaptively aggregate useful frequencies.

## 3 METHOD

### 3.1 Model Formulation

A noise model is given by $Y = X + N$, where $Y$, $X$, and $N$ ($\in \mathbb{R}^{H \times W}$) denote a noisy image, a clean image, and the additive noise component, respectively. The channel dimension is omitted for brevity. Let $G \in \mathbb{R}^{H \times W}$ represent a clean guidance image, due to the high structure similarity between $G$ and $X$, we can use $G$ to help remove the noise and compensate for the missing details. However, directly fusing $G$ and $Y$ will introduce undesired artifacts due to inconsistency. Therefore, we perform calibration on the cross-spectral features before fusion. For the $k$-th channel of guidance features, $g_k$, we first obtain a weight matrix by a transformation $A_k(\cdot, \cdot)$, which establishes the correlation between $y_k$ and $g_k$ to eliminate inconsistency by:

$$\tilde{g}_k = A_k(g_k, y_k) \cdot g_k, \tag{1}$$

where $\tilde{g}_k$ and $y_k$ represent the $k$-th channel of calibrated guidance features and noisy features, respectively. We can obtain the calibrated noisy features $\tilde{y}_k$ by applying the same operation. Then, the predicted clean image $\hat{Y} \in \mathbb{R}^{H \times W}$ can be produced by:

$$\hat{Y} = \sum_k (B_k(\tilde{y}_k, \tilde{g}_k) \cdot \tilde{y}_k) * f_k, \tag{2}$$

where $B_k(\cdot, \cdot)$ is another transformation which establishes the correlation between $\tilde{y}_k$ and $\tilde{g}_k$, focusing on compensating for missing texture details and smoothing the flat areas. The symbol $*$ denotes the convolutional operation, and $\{f_k\}_{k=1}^{K} \in \mathbb{R}^{H \times W \times K}$ is the feature filters.

Considering the different roles of high-frequency edges and low-frequency smoothing areas in guided denoising, we decouple $B_k$ into $B_k^h$ and $B_k^l$ to selectively capture useful high-/low-frequency signals from the guidance features and highlight the informative frequency components. As a result, Eq. 2 can be rewritten as:

$$\hat{Y} = \sum_k ((B_k^h(\tilde{y}_k, \tilde{g}_k) + B_k^l(\tilde{y}_k, \tilde{g}_k)) \cdot \tilde{y}_k) * f_k, \tag{3}$$

where $B_k^h$ and $B_k^l$ are transformations responsible for high-frequency correlation and low-frequency correlation, respectively. Overall, our model is built based on Eq. 1 and Eq. 3, where $A$, $B$ are implemented by our Multi-Scale Inconsistency Calibration Module (MICM) and Dual-Domain Adaptive Fusion Module (DAFM), respectively.

In addition, we introduce a pre-denoising stage before the above operations to reduce the impact of noise on the calibration and fusion processes.

### 3.2 Model Architecture

As illustrated in Fig. 2, our RFFNet is an end-to-end two-stage network for progressively restoring noise-free and detail-rich images. The first stage performs pre-denoising to not only facilitate noise removal in the fusion stage but also provide the general outline for easily identifying inconsistency. The guidance features are involved in the network based on our MICM and DAFM in the second stage. A Supervised Attention Module (SAM) [44] is used to bridge the features from the first stage to the second one for further processing.

Both stages comprise an encoder-decoder architecture with two downsampling layers and two upsampling layers. A skip connection followed by a 3x3 convolution is employed between the encoder and decoder features. A ResBlock consists of two 3x3 convolution layers, each followed by a PReLU [19].

### 3.3 Multi-Scale Inconsistency Calibration Module

Our MICM addresses inconsistency by progressively calibrating the cross-spectral features in a coarse-to-fine manner. As illustrated in Fig. 2, given the guidance feature $G \in \mathbb{R}^{C \times H \times W}$ and pre-denoised feature $N \in \mathbb{R}^{C \times H \times W}$, we leverage average pooling (AP) with different downsampling rates to transform them into different spaces. Here, AP can serve as a low-pass filter, which significantly reduces the impact of noise at lower scales, and achieves effective receptive fields for large-scale inconsistency.

In the three branches of the downsampling part, we apply a spatial attention module (SA) to the cross-modal features to calibrate the inconsistency. To be specific, SA leverages a simple convolutional layer to fuse the cross features and distributes them to distinct convolutional layers for generating attention weights adaptively. The process of SA is expressed as:

$$Z = \text{ReLU}\left(Conv([N, G])\right),$$
$$\hat{G} = G \cdot \mathcal{S}(Conv_G(Z)), \ \hat{N} = N \cdot \mathcal{S}(Conv_N(Z)), \tag{4}$$

$[\cdot, \cdot]$ denotes the concatenation operation, $\mathcal{S}$ is the sigmoid operation. Through SA, we can obtain the reweighted features for mitigating the inconsistency.

Then, the calibrated features are upsampled to the original size and aggregated with the initial features through skip connections, followed by a convolutional layer for summarizing. The calibration process is applied to both noisy features and guidance features. For brevity, we provide a concise description of the calibration process for the guidance features:

$$\hat{G}_i = \text{SA}(AP_{2^{4-i}}(G) + \hat{G}_{i-1} \uparrow_2, \ AP_{2^{4-i}}(N) + \hat{N}_{i-1} \uparrow_2),$$
$$\hat{G} = Conv(\sum_i (\hat{G}_i \uparrow_{2^{4-i}}) + G), \tag{5}$$

where, $i \in \{1, 2, 3\}$, $\hat{G}_0 = 0$, $AP_{2^{4-i}}$ represents the average pooling operation with a downsampling rate as $2^{4-i}$, and $\uparrow_2$ denotes the *bilinear* interpolation operation with an upsampling rate as 2. The calibrated noise features $\tilde{N}$ yield through another branch of the same structure. Next, $\tilde{G}$ and $\tilde{N}$ are injected into the main path.

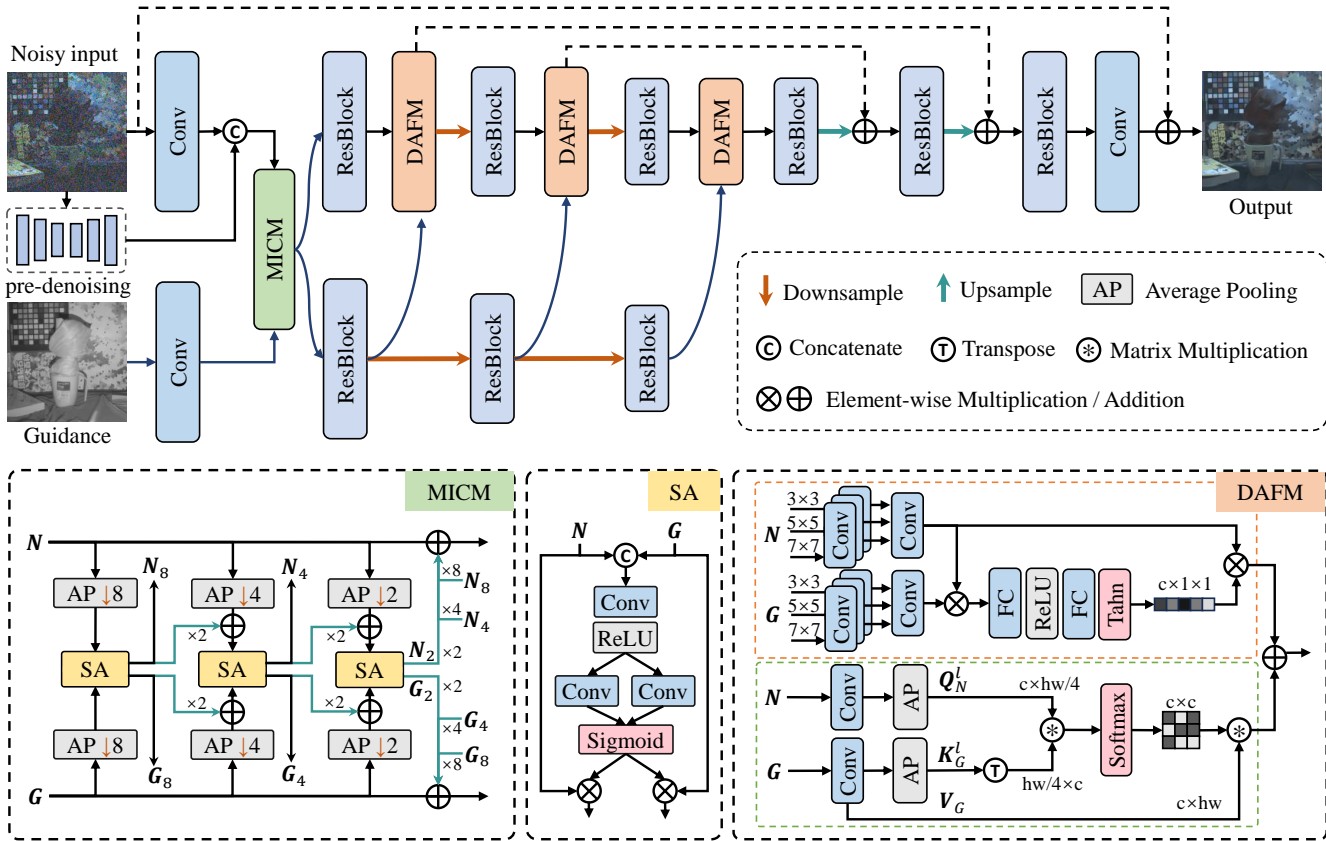

**Figure 2: Overall architecture of the proposed RFFNet. The first stage performs pre-denoising and the second stage executes fusion. Multi-Scale Inconsistency Calibration Module (MICM) calibrates the cross-spectral features before fusion. Each encoding layer of the fusion stage ends with a Dual-Domain Adaptive Fusion Module (DAFM), which selectively captures different informative frequencies for high-quality denoising.**

### 3.4 Dual-Domain Adaptive Fusion Module

To extract different frequencies of the features, a straightforward way is to utilize wavelet transform or Fourier transform. However, it is difficult for these tools to distinguish the frequency components to be enhanced or suppressed and they require additional computational overhead to transform features back to the spatial domain [9]. Instead, we use adaptive frequency filters [12] to yield high/low input frequencies. For high-frequency extraction, we divide features into several groups and apply depthwise convolutions with different kernel sizes to each group to simulate the cutoff frequencies in different high-pass filters. For low-frequency extraction, we use average pooling as low-pass filters. We apply the frequency extraction above to both noisy features and guidance features. Then, high/low frequencies are fed into the local and global branches respectively to accentuate the informative signals.

**High-frequency local branch**. Adaptive high-pass filters with different kernel sizes possess distinct receptive fields, enabling the differential extraction of high-frequency components and the provision of varied local information. We employ dynamic high-pass filters on both $N$ and $G$. For the $i$-th group of $N$, $N_i$, we can

obtain the corresponding high frequencies $N_i^h$ via

$$N_i^h = \mathcal{H}_k(N_i), \tag{6}$$

Here, $\mathcal{H}_k$ represents the depth-wise convolution with kernel size $k \times k$, $k \in \{3, 5, 7\}$. The high-frequency features $N_h$ and $G_h$ are yielded by concatenating all groups. Then we multiply $N_h$ and $G_h$ and fed the result into a modified squeeze-and-excitation block to obtain a local attention map as

$$A = \text{Tahn}(W_2(\text{ReLU}(W_1(N_h * G_h)))), \tag{7}$$

Where $W_1$ and $W_2$ denote parameters of the fully connected layers. We utilize the Tanh function instead of the sigmoid function, since Tanh projects attention weights into $(-1, 1)$, the negative weights can help suppress the detrimental frequencies. The resulting attention map is subsequently multiplied with the initial high frequencies $G_h$. Then we can obtain the high-frequency output.

**Low-frequency global branch**. We use channel-wise cross attention to capture the useful global low frequencies, where the queries ($Q$) come from the noise branch and the keys ($K$) and values ($V$) come from the guidance branch. The attention map is generated

**Table 1: The average PSNR, SSIM, and LPIPS on the DVD [21] test set with noise level $\sigma \in \{2, 4, 6\}$. The best and second-best results are highlighted in boldface and underlined, respectively.**

| Methods | $\sigma$=2 | | | $\sigma$=4 | | | $\sigma$=6 | | |
|---|---|---|---|---|---|---|---|---|---|
| | PSNR↑ | SSIM↑ | LPIPS↓ | PSNR↑ | SSIM↑ | LPIPS↓ | PSNR↑ | SSIM↑ | LPIPS↓ |
| RIDNet [2] | 30.31 | 0.937 | 0.083 | 28.16 | 0.908 | 0.118 | 26.60 | 0.884 | 0.143 |
| SADNet [4] | 30.69 | 0.936 | 0.082 | 28.87 | 0.924 | 0.101 | 27.24 | 0.903 | 0.126 |
| NBNet [7] | 31.38 | 0.948 | 0.073 | 29.14 | 0.926 | 0.100 | 27.27 | 0.906 | 0.124 |
| MPRNet [44] | 31.79 | 0.950 | 0.068 | 29.37 | 0.928 | 0.099 | 27.68 | 0.908 | 0.123 |
| Restormer [43] | 31.42 | 0.950 | 0.067 | 29.11 | 0.927 | 0.099 | 27.69 | 0.909 | 0.121 |
| NAFNet [6] | 31.51 | 0.949 | 0.065 | 29.28 | 0.928 | 0.098 | 27.49 | 0.908 | 0.122 |
| DKN [22] | 27.22 | 0.890 | 0.115 | 24.34 | 0.843 | 0.139 | 22.78 | 0.807 | 0.157 |
| SVLRM [32] | 27.70 | 0.900 | 0.098 | 25.29 | 0.857 | 0.129 | 23.43 | 0.823 | 0.149 |
| CUNet [11] | 28.92 | 0.924 | 0.090 | 27.24 | 0.901 | 0.111 | 26.01 | 0.878 | 0.127 |
| SANet [35] | 30.04 | 0.938 | 0.0750 | 27.83 | 0.917 | 0.098 | 26.41 | 0.901 | 0.111 |
| DVN [21] | 31.50 | 0.955 | 0.058 | 29.62 | 0.940 | 0.079 | 28.26 | 0.927 | 0.095 |
| **RFFNet** | **32.22** | **0.962** | **0.053** | **30.20** | **0.949** | **0.075** | **28.69** | **0.937** | **0.090** |

by multiplying $Q$ and $K$. Our low-frequency filters with a downsampling rate are applied to $Q$ and $K$, to learn global representations in the low-frequency space and reduce computational complexity simultaneously. The resulting channel-wise attention map is then multiplied with $V$. We can obtain the low-frequency output via

$$F_l = \text{Softmax}\left(\mathcal{P}(Q_N)\mathcal{P}(K_G)^T / \sqrt{d_k}\right) V_G, \tag{8}$$

where $Q_N \in \mathbb{R}^{C \times HW}$, $K_G \in \mathbb{R}^{C \times HW}$ is generated by applying $1 \times 1$ convolutions to the $N$ and $G$ respectively, while $V_G \in \mathbb{R}^{C \times HW}$ is produced from $G$ with $1 \times 1$ convolution. $\mathcal{P}$ denotes the average pooling operation with a downsampling rate of 2. The channel-wise CA can also provide lower complexity than the spatial version. In addition, we deploy the multi-head mechanism [37] to further save the complexity and enhance the diversity of feature spaces.

Finally, we integrate the initial noisy features with the outputs of the two branches, then feed the result into the DFFN [24] for further frequency refinement.

## 3.5 Loss Function

We adopt the Charbonnier loss [5] in both spatial and frequency domains to facilitate dual-domain learning:

$$\mathcal{L}_s = \mathcal{L}_{charbonnier}(\hat{X}_1, Y) + \mathcal{L}_{charbonnier}(\hat{X}_2, Y), \tag{9}$$

$$\mathcal{L}_f = \mathcal{L}_{charbonnier}(\mathcal{F}(\hat{X}_1), \mathcal{F}(Y)), \tag{10}$$

where $\hat{X}_1$ and $\hat{X}_2$ are the outputs of the pre-denoising sub-network and the whole network, respectively; $Y$ is the reference; and $\mathcal{F}$ is the fast Fourier transform. The final loss function is given by $\mathcal{L} = \mathcal{L}_s + \lambda \mathcal{L}_f$, where $\lambda$ is set as 0.1.

## 4 EXPERIMENTS

In this section, we introduce the implementation details of training RFFNet and present experimental results on NIR-guided RGB image denoising and flash-guided no-flash image denoising to illustrate the effectiveness of our model.

### 4.1 Experimental settings

*Datasets and Metrics.* NIR-guided RGB image denoising is evaluated on the DVD [21]. It consists of 307 high-resolution 10-bit raw data, which are converted to 5k RGB-NIR image pairs of size $3 \times 256 \times 256$ used for training, and 1k image pairs ($3 \times 256 \times 256$) used for testing. Flash-guided no-flash image denoising is evaluated on the FAID [1], which includes 2775 flash/no-flash aligned pairs categorized into six classes: people, shelves, plants, toys, rooms, and objects. Following [31], we randomly select 256 pairs for validation and 256 pairs for testing and use the rest for training. In terms of evaluation metrics, PSNR, SSIM, and LPIPS [46] are used.

*Training Details.* The proposed network is trained on $128 \times 128$ patches using the Adam optimizer [23] ($\beta_1 = 0.9$, $\beta_2 = 0.999$) with a batch size of 16. The augmentation strategy follows that of [44], including random flipping and rotating. The initial learning rate is set to $2e^{-4}$, which is gradually reduced to $1e^{-6}$ with cosine annealing [30]. For NIR-guided RGB image denoising, we follow the settings in [21], where we randomly reduce the average value of raw images to simulate low-light condition and then add the Gaussian-Poisson mixed noise with $\sigma$ ranging from 1 to 16 to the pseudo-dark raw images. The model is trained for 80 epochs. For flash-guided no-flash image denoising, we randomly add Gaussian noise with noise levels ranging from 10 to 100 to the reference image. The training process takes 1500 epochs. All models are implemented on an NVIDIA Tesla A100 GPU using PyTorch.

### 4.2 Experimental Results

**NIR-guided RGB image denoising.** We evaluate RFFNet on the DVD [21] test set under Gaussian-Poisson mixed noise with $\sigma \in \{2, 4, 6\}$. We compare our results with state-of-the-art guided image denoising approaches, including DKN [22], SVLRM [32], CUNet [11], SANet [35] and DVN [21]. Additionally, we also consider single-image denoising methods, such as RIDNet [2], SADNet [4], NBNet [7], MPRNet [44], Restormer [43], and NAFNet [6]. All compared methods are trained on the same training set as ours.

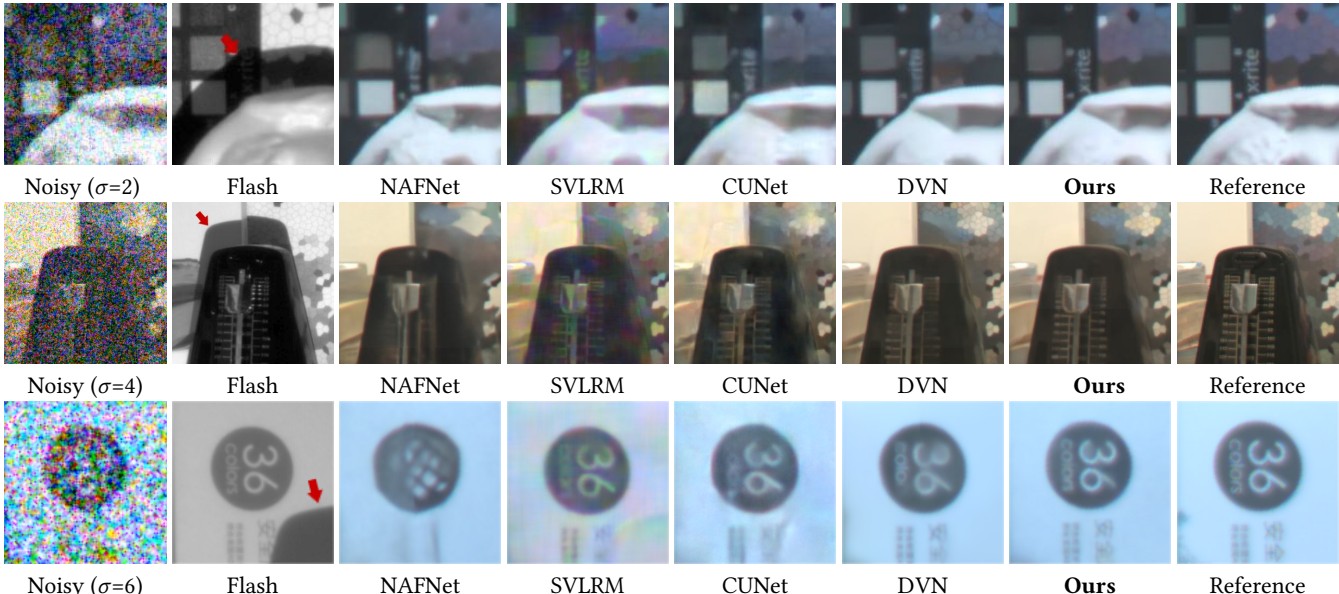

**Figure 3: The qualitative comparison among our RFFNet and the state-of-the-art methods on the noisy RGB-NIR pairs from DVD [21] with different noise levels. Images are brightened for display.**

**Table 2: The average PSNR, SSIM, and LPIPS on the FAID [1] test set with noise level $\sigma \in \{25, 50, 75\}$. The best and second-best results are highlighted in boldface and underlined, respectively.**

| Methods | $\sigma$=25 | | | $\sigma$=50 | | | $\sigma$=75 | | |
|---|---|---|---|---|---|---|---|---|---|
| | PSNR↑ | SSIM↑ | LPIPS↓ | PSNR↑ | SSIM↑ | LPIPS↓ | PSNR↑ | SSIM↑ | LPIPS↓ |
| RIDNet [2] | 35.86 | 0.966 | 0.286 | 33.06 | 0.942 | 0.345 | 31.31 | 0.922 | 0.374 |
| SADNet [4] | 36.32 | 0.969 | 0.269 | 33.79 | 0.950 | 0.330 | 32.19 | 0.934 | 0.364 |
| NBNet [7] | 36.31 | 0.969 | 0.277 | 33.67 | 0.949 | 0.336 | 32.04 | 0.932 | 0.371 |
| MPRNet [44] | 36.65 | 0.970 | 0.275 | 34.07 | 0.952 | 0.331 | 32.46 | 0.937 | 0.364 |
| Restormer [43] | 36.68 | 0.970 | 0.272 | 34.15 | 0.952 | 0.326 | 32.56 | 0.937 | 0.358 |
| NAFNet [6] | 36.74 | 0.971 | 0.263 | 34.24 | 0.953 | 0.315 | 32.69 | 0.939 | 0.346 |
| DKN [22] | 34.94 | 0.958 | 0.271 | 32.46 | 0.941 | 0.298 | 30.82 | 0.924 | 0.330 |
| SVLRM [32] | 35.22 | 0.962 | 0.268 | 32.97 | 0.945 | 0.294 | 31.34 | 0.930 | 0.325 |
| CUNet [11] | 36.21 | 0.970 | 0.241 | 34.14 | 0.956 | 0.277 | 32.72 | 0.945 | 0.294 |
| SANet [35] | 36.07 | 0.969 | 0.262 | 34.07 | 0.955 | 0.285 | 32.81 | 0.946 | 0.297 |
| DeepFnF [39] | 36.83 | 0.973 | 0.238 | 35.01 | 0.963 | 0.264 | 33.84 | 0.956 | 0.277 |
| RIDFnF [31] | 36.98 | 0.974 | 0.237 | 35.19 | 0.964 | 0.263 | 33.95 | 0.957 | 0.276 |
| RFFNet-s | 37.42 | 0.975 | 0.236 | 35.61 | 0.967 | 0.261 | 34.43 | 0.960 | 0.276 |
| RFFNet | **37.84** | **0.977** | **0.230** | **36.06** | **0.969** | **0.256** | **34.93** | **0.963** | **0.267** |

Tab. 1 shows that our method achieves the best results across three noise levels.

The qualitative comparisons in Fig. 3 clearly illustrate the significant advantages of our method over other state-of-the-art approaches on noise removal, artifact suppression, and detail recovery. For single-image denoisers like NAFNet, have achieved satisfactory results on weak-noise datasets. However, as the synthetic noise intensity increases, NAFNet has to sacrifice a large amount of edges and details. For the guided denoising methods, SVLRM and CUNet inevitably yield partially denoised results when confronted with

strong noise. Moreover, they fail to handle the inconsistency between RGB and NIR image pairs, resulting in severe artifacts. DVN effectively removes noise but fails to estimate the complete structure of the target image, consequently, the details of the NIR image cannot be fully integrated into the RGB image.

By contrast, our RFFNet exhibits strong denoising capabilities and effectively suppresses artifacts. Furthermore, we extract useful frequencies from the NIR image and selectively aggregate edges and textures, yielding results with rich details.

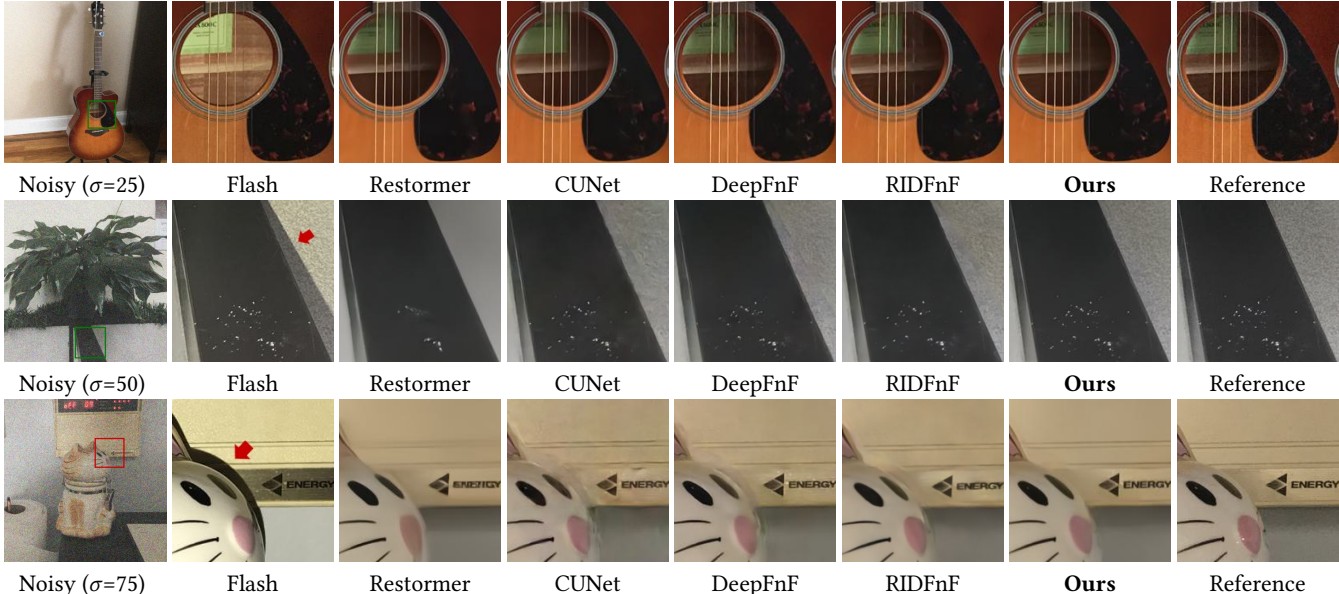

**Figure 4: The qualitative comparison among our RFFNet and the state-of-the-art methods on the flash/no-flash image pairs from FAID [39] with different noise levels.**

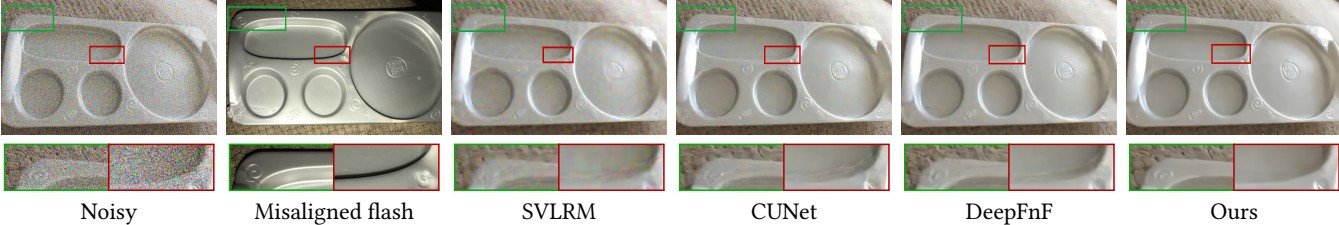

**Figure 5: Visual comparisons on a misaligned flash/no-flash image pair. Our method results in fewer artifacts and better details.**

**Flash-guided no-flash image denoising.** We perform validation on the test set of FAID [1] under Gaussian noise with $\sigma \in \{25, 50, 75\}$. For this task, in addition to the methods mentioned above, we compare our RFFNet with DeepFnF [39] and RIDFnF [31], which are elaborately designed for flash-guided denoising. To further demonstrate the superiority of our method, we additionally provide the flash-guided denoising results of a small version (-s) by changing the channel count of three scales from {64, 96, 128} to {32, 44, 56}, which has only 0.99 M parameters. Tab. 2 shows that our RFFNet, even RFFNet-s, outperforms all comparative methods in terms of PSNR, SSIM, and LPIPS under different noise levels.

Fig. 4 shows that Restormer over-smooth the details. Guided denoising methods, including CUNet, DeepFnF, and RIDFnF, produce ghosting artifacts in the bottom two images due to the shadows in the flash image. By contrast, RFFNet completely removes noise and preserves fine structures faithful to the ground truth. Moreover, as shown in Fig. 5, our method yields fewer artifacts and better details compared to other fusion-based methods in the misaligned case.

Besides, as listed in Tab. 3, both RFFNet and RFFNet-s demonstrate computational efficiency and possess a smaller model size compared to the state-of-the-art denoisers.

**Real-world low-light image denoising.** We conduct qualitative experiments on RGB-NIR pairs captured in real low-light environments. As shown in Fig. 6, the details of the RGB image (brightened for display) are obscured by dense noise, and the NIR image exhibits significant highlights and shadows. Restormer introduces a large amount of artifacts. Both CUNet and DVN struggle to address artifact suppression and denoising. Our method effectively eliminates inconsistency and introduces richer details.

### 4.3 Ablation Studies

We conduct ablation studies with RFFNet on the FAID dataset [1] under $\sigma = 75$ to investigate the influences of our proposed components. The results of the MICM and DAFM are presented in Tab. 4.

**Influence of each component.** For the network design, we compare the 2-stage architecture, MICM, DAFM, the frequency loss, and the full model with the baseline model. Specifically, for the baseline, we utilize a single U-Net to be the backbone and apply addition operation for fusion in multi-level encoders. By deploying MICM, the model receives a performance gain of 0.29 dB (Tab. 4 (b)). A pre-denoising stage is added to the model for progressive denoising

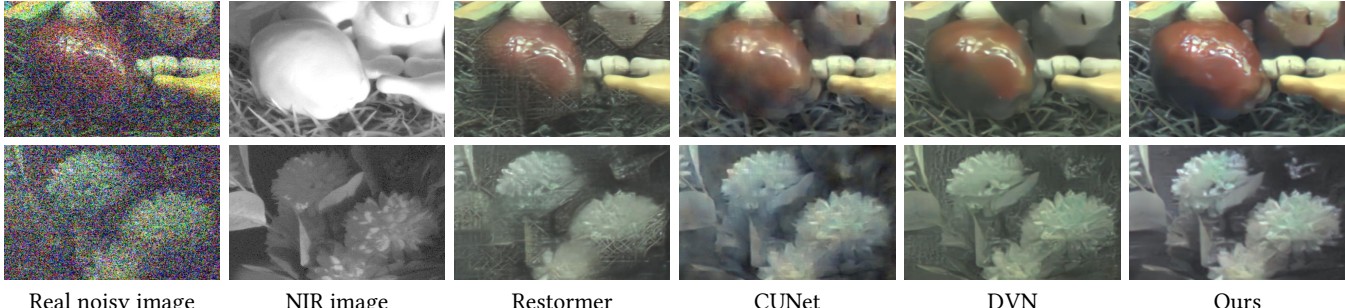

| Real noisy image | NIR image | Restormer | CUNet | DVN | Ours |

**Figure 6: Visual comparisons on real-world RGB/NIR image pairs. Our method results in fewer artifacts and better details.**

**Table 3: Comparisons of computation overhead. Compared to state-of-the-art methods based on patches of size $128 \times 128$, RFFNet and RFFNet-s achieve superior efficiency and possess smaller model sizes.**

| Method | Restormer [43] | NAFNet [6] | DVN [21] | CUNet [11] | SANet [35] | DeepFnF [39] | RIDFnF [31] | RFFNet | RFFNet-s |
|---|---|---|---|---|---|---|---|---|---|
| Params/M | 26.10 | 115.86 | 6.96 | 0.69 | 4.64 | 78.47 | 1.84 | 4.57 | 0.99 |
| FLOPs/G | 35.25 | 15.83 | 57.78 | 10.1 | 26.47 | 14.01 | 60.32 | 22.62 | 5.84 |
| Time/ms | 37.4 | 25.2 | 12.4 | 4.81 | 40.98 | 11.7 | 16.4 | 11.4 | 9.8 |
| PSNR/dB | 36.68 | 36.74 | - | 36.21 | 36.07 | 36.83 | 36.98 | 37.84 | 37.42 |

**Table 4: Ablation study of various components of our method on FAID [1] ($\sigma$=75).**

| | Baseline | 2-stage | MICM | DAFM | $\mathcal{L}_f$ | PSNR | SSIM | LPIPS |
|---|---|---|---|---|---|---|---|---|
| (a) | ✓ | | | | | 33.97 | 0.956 | 0.280 |
| (b) | ✓ | | | ✓ | | 34.26 | 0.958 | 0.280 |
| (c) | ✓ | ✓ | ✓ | | | 34.53 | 0.960 | 0.275 |
| (d) | ✓ | ✓ | ✓ | ✓ | | 34.82 | 0.962 | 0.271 |
| (e) | ✓ | ✓ | ✓ | ✓ | ✓ | **34.93** | **0.963** | **0.267** |

**Table 5: Ablation on the branches of MICM.**

| Branch | PSNR | Params | FLOPs |
|---|---|---|---|
| ↓2 | 34.79 | 4.13 | 22.34 |
| ↓2,4 | 34.87 | 4.35 | 22.56 |
| ↓2,4,8 | 34,93 | 4.57 | 22.62 |

**Table 6: Ablation on different fusion strategies.**

| Method | PSNR | Params | FLOPs |
|---|---|---|---|
| W-MCA | 31.81 | 4.46 | 22.37 |
| C-MCA | 34.80 | 4.46 | 22.36 |
| DAFM | 34.93 | 4.57 | 22.62 |

and facilitating the calibration process, leading to a 0.27 dB improvement (Tab. 4 (c)). Then, we apply DAFM instead of addition, the model improves by 0.29 dB (Tab. 4 (d)). To facilitate the frequency learning, we introduce the frequency loss $\mathcal{L}_f$ and obtain a gain of 0.09 dB (Tab. 4 (e)). Finally, all our contributions together yield a substantial improvement of 0.94 dB in PSNR over the baseline, setting the new state-of-the-art in low-light image denoising.

**The Effectiveness of MICM.** As shown in the example of Fig. 7, the full model with MICM handles the inconsistent regions better. To further demonstrate the effectiveness of the coarse-to-fine manner, we do experiments by progressively adding downsampling branches of MICM, leading to higher scores (Tab. 5).

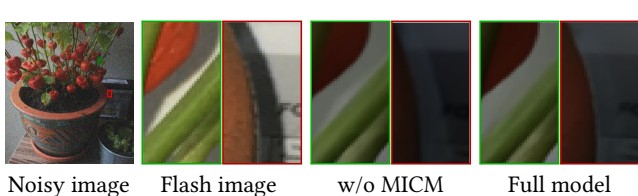

| Noisy image | Flash image | w/o MICM | Full model |

**Figure 7: Visual comparison on a flash/no-flash image pair existing inconsistency.**

**Compare with other fusion strategies.** We compare DAFM with typical fusion methods such as window multi-head cross attention (W-MCA) and channel-wise multi-head cross attention (C-MCA). DAFM achieves a minimum improvement of 0.12 dB while increasing the parameters by only 2.5% (+0.11 M).

## 5 CONCLUSION

In this paper, we propose a two-stage framework, dubbed RFFNet, for low-light image denoising based on a robust and flexible fusion strategy. Specifically, our Multi-Scale Inconsistency Calibration Module transfers cross-modal features into multi-scale spaces and applies spatial attention in each branch to calibrate inconsistency in a coarse-to-fine manner, ensuring the robustness of the fusion process. Moreover, we develop a Dual-Domain Adaptive Fusion Module to excavate more information from the guidance image by extracting high/low frequencies from the cross-spectral features and adaptively emphasizing the informative frequency components. Extensive experiments on NIR-guided RGB image denoising and flash-guided no-flash image denoising demonstrate that our method outperforms state-of-the-art algorithms in terms of noise removal, artifact suppression, and detail recovery.

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
