# OpenReview forum: "RFFNet: Towards Robust and Flexible Fusion for Low-Light Image Denoising"
_acmmm.org/ACMMM/2024/Conference — MM2024 Poster_

### Official Review · Reviewer_sZmq · 2024-05-14

**Rating:** 3
**Confidence:** 4

**Summary:**

The authors of this paper propose a robust and flexible vision network (RFFNet) for low light image denoising. Specifically, a multi-scale inconsistency calibration module is proposed to process the inconsistent features into the multi-scale space before fusion by first mapping guidance, and to pre-denoise the features in a coarse-to-fine manner. In addition, by designing a dual-domain adaptive fusion module, adaptive extraction of useful high/low frequency signals from guidance characteristics and then highlighting information frequencies is realized. A large number of experimental results verify the effectiveness of the method.

**Strengths:**

The main contributions of the paper include the proposed multi-scale inconsistencies calibration module to deal with local inconsistencies, and the design of a two-domain adaptive fusion module to adaptively fuse information.
The experimental part shows the advantages of RFFNet in visual effects and performance indicators , as well as in computational efficiency.

**Limitations:**

1. The formula modeling in this paper is incomplete. In formula 1, only additive noise is considered without multiplicative noise, which is not consistent with the noise distribution in the real low-light scene.

2. The setting of this paper is simple and does not conform to the real situation. In the setup of this paper, it is considered that the guided NIR or flash image is perfectly aligned with the GT image, which is a very simple setup. However, in the real world, most of the guide images should not be perfectly aligned. For this more complex setup, the authors present only a visual comparison result that lacks a GT image and does not show any metric results, which is unconvincing.

3. The experimental exposition of this paper is not clear. For the experiments on FAID dataset, the author did not fully explain whether the data used was RAW or RGB, and it seems that the adding  noise method on FAID dataset is not same with that on DVD dataset or other compared methods such as deepFnF, which does not meet the setting of dark light image denoising. Authors are advised to fully explain the practices and the reasons for the differences.

4. This paper does not seem to have strong relevance to the MM conference. Specifically, there is no MM application or context, no actual MM results, no MM-related references, and even no mention of multimedia in the full text.

**Suitability:**

1

---

### Official Review · Reviewer_sqTT · 2024-05-22

**Rating:** 4
**Confidence:** 2

**Summary:**

The RFFNet paper addresses a crucial challenge in low-light image denoising by introducing a novel fusion network that leverages guidance images to enhance noise reduction and detail preservation.

**Strengths:**

1.  The paper proposes a unique approach that integrates a Multi-Scale Inconsistency Calibration Module (MICM) and a Dual-Domain Adaptive Fusion Module (DAFM) to tackle inconsistencies in guided image denoising.
2. The paper reports superior performance over current state-of-the-art methods on both NIR-guided and flash-guided denoising tasks.
3. The architecture cleverly addresses both high and low-frequency components in the image denoising process, potentially setting a new benchmark in the field.

**Limitations:**

1. There are many grammar problems, typos, and format errors.
2. The contributions of this work seem incremental.
3. It’s better to add some visual results to verify the contribution of each loss function term more intuitively and clearly.
4. Discuss the limitations of the proposed method more thoroughly, including potential failure cases, and suggest future research directions to overcome these limitations.

**Suitability:**

2

---

### Official Review · Reviewer_QTJx · 2024-05-24

**Rating:** 4
**Confidence:** 3

**Summary:**

This paper proposes a fusion network for low-light image denoising, and a multi-scale inconsistency calibration module is introduced to guide the feature calibration in a coarse-to-fine manner. The work also develops a dual-domain adaptive fusion module to extract useful information from the guidance features. experimental results demonstrate the effectiveness on NIR-guided RGB image denoising and flash-guided no flash image denoising.

**Strengths:**

1. The writing of this paper is easy to understand.
2. The experiments are sufficient and the visual results and quantitative metrics verify the effectiveness of this method.
3. The description of the methodology is detailed.

**Limitations:**

1. The motivation of the paper requires more detailed argumentation, such as whether the approach for extracting high and low frequencies is reasonable in DAFM.

**Suitability:**

2

---

### Official Review · Reviewer_xf2s · 2024-05-24

**Rating:** 4
**Confidence:** 3

**Summary:**

This paper presents RFFNet, a two-stage framework for low-light image denoising, featuring a robust and flexible fusion strategy. The Multi-Scale Inconsistency Calibration Module (MICM) transfers cross-modal features into multi-scale spaces, employing spatial attention to calibrate inconsistency in a coarse-to-fine manner. Additionally, the Dual-Domain Adaptive Fusion Module (DAFM) extracts high/low frequencies from cross-spectral features, emphasizing informative frequency components adaptively. Experimental results on NIR-guided RGB image denoising and flash-guided no-flash image denoising showcase superior performance in noise removal, artifact suppression, and detail recovery compared to state-of-the-art algorithms.

**Strengths:**

1. This paper demonstrates good readability, with clear problem analysis and a well-structured, logically coherent solution approach.
2. The paper tackles the fusion of noisy low-light images with guide images in an interesting and novel manner. While the use of well-known attention modules in RFFNet somewhat limits the novelty in module design, the adjustments made to these modules reflect the authors' deep understanding of guided image restoration. I appreciate these adjustments and think that it maches the tone of ACM Multimedia.
3. The experimental results presented in this paper are impressive. There are significant advantages in terms of quantitative metrics, visual effects, and computational overhead.

**Limitations:**

I found the ablation study in Table 4 to be quite confusing. Is it reasonable for Unet to achieve a PSNR of 33.97dB as a baseline? This seems somewhat high and deviates from common expectations. Would retraining other methods using the training schedule proposed in this paper lead to different conclusions?
This observation undermines my trust in the fairness of comparisons in this paper. If the authors could address my concerns, I would be inclined to give a higher rating.

By the way, the template of this paper is strange. I'm not sure if it should be considered when rating.

**Suitability:**

3

---

### Meta-Review · Area_Chair_91Ga · 2024-07-02

**Recommendation:** Accept (Poster)
**Confidence:** 5

**Metareview:**

All reviewers have decided to accept this work. AC agrees with each reviewer and decides to accept this paper. The authors are suggested to consider all suggestions in the camera ready.